# Limited response of primary nasal epithelial cells to *Bordetella pertussis* infection

Martin Zmuda,[1] Ivana Malcova,[1] Barbora Pravdova,[1] Ondrej Cerny,[1] Denisa Vondrova,[1] Jana Kamanova[1]

**ABSTRACT** *Bordetella pertussis* is a Gram-negative coccobacillus that causes whooping cough or pertussis, a respiratory disease that has recently experienced a resurgence. Upon entering the respiratory tract, *B. pertussis* colonizes the airway epithelium and attaches to ciliated cells. Here, we used primary human nasal epithelial cells (hNECs) cultured at the air-liquid interface and investigated their interaction with *B. pertussis* B1917, focusing on the role of the type III secretion system effector protein BteA. In this model, which resembles the epithelial cells of nasal epithelium *in vivo*, *B. pertussis* B1917 localized predominantly in the overlying mucus and scarcely colonized the cell cilia. The colonization led to a gradual decline in epithelial barrier function, as shown by measurements of transepithelial electrical resistance (TEER) and staining of the tight junction protein zonula occludens 1. The decrease in TEER occurred independently of the cytotoxic effector protein BteA. Transcriptomic and proteomic analyses of hNECs showed only moderate changes following infection, primarily characterized by increased mucus production, including upregulation of mucin MUC5AC. No profound response to BteA was detected. Furthermore, the infection did not induce production of inflammatory cytokines, suggesting that *B. pertussis* B1917 evades recognition by hNECs in this model system. These results suggest that the mucus may serve as a niche that allows *B. pertussis* B1917 to minimize epithelial recognition and damage. The lack of a robust immune response further indicates that additional components of the nasal mucosa, such as innate immune cells, are likely required to initiate an effective host defense.

**IMPORTANCE** The nasal epithelium is the initial site where *Bordetella pertussis* comes into contact with the host during respiratory tract infection. In this study, human nasal epithelial cells cultured at the air-liquid interface were established as an *in vitro* model to investigate the early stages of *B. pertussis* infection. We showed that the clinical isolate *B. pertussis* B1917 resides in the mucus during the early stages of colonization without disrupting the epithelial barrier function. Infection results in moderate transcriptomic and proteomic changes, characterized by increased mucus production and minimal inflammatory signaling. These results suggest that *B. pertussis* B1917 may evade early host recognition by residing in mucus and avoiding direct interaction with epithelial cells. They also highlight the importance of other components of the mucosal immune system, such as resident immune cells, for the initiation of an effective defense.

**KEYWORDS** *Bordetella pertussis*, airway epithelium, air-liquid interface culture, human nasal epithelial cell, type III secretion system, BteA effector

*B*ordetella pertussis, a Gram-negative coccobacillus, is the causative agent of whooping cough or pertussis. This highly transmissible respiratory disease presents as an acute, often fatal illness in infants and as a persistent, severe paroxysmal cough with characteristic whooping sounds in adults (1). Pertussis continues to be a signifi-cant health concern, and it was estimated that in 2014 there were 24 million cases in

Address correspondence to Jana Kamanova, kamanova@biomed.cas.cz.

The authors declare no conflict of interest.

See the funding table on p. 20.

children under 5 years with 160,000 deaths (2). While the highest incidence is observed in unvaccinated populations, a resurgence of the disease has been seen in developed countries that have transitioned from whole-cell pertussis (wP) vaccines to less reactive acellular pertussis (aP) vaccines (3). Current aP vaccines have been demonstrated to be insufficient in preventing nasal colonization by *B. pertussis* (4). This is likely due to their restricted ability to elicit strong mucosal immunity, including induction of secretory IgA as well as IL-17 and IFN-γ-producing respiratory tissue-resident memory T cells ($T_{RM}$) (5–8). As a result, subclinical infections with *B. pertussis* are relatively common in aP-vaccinated children and adolescents, which poses a considerable risk of transmission to unvaccinated or incompletely vaccinated infants (9). Recently, several developed countries, including the Czech Republic, have reported outbreaks of pertussis. This resurgence may be related to the reduced circulation of *B. pertussis* during the COVID-19 pandemic, which led to increased susceptibility of the population (10).

*B. pertussis* is a pathogen exclusively adapted to humans with no animal or environmental reservoirs, which is transmitted via respiratory droplets. After inhalation, *B. pertussis* has been reported to attach to the ciliated cells and colonize the ciliated pseudostratified columnar epithelium of the respiratory tract (11, 12). This epithelium consists of ciliated cells and secretory club and goblet cells, which are essential for mucociliary clearance and barrier function. Besides, multipotent basal cells are responsible for epithelial regeneration (13, 14). Advances in single-cell transcriptomics have further identified rarer cell types within the airway epithelium, including tuft cells, pulmonary neuroendocrine cells, ionocytes, and various transitional and intermediate cell types (13–16). The airway epithelium actively contributes to respiratory defense by secreting antimicrobial substances and interacts with stromal, neuronal, and immune cells (both tissue-resident and recruited) to maintain homeostasis and protect the respiratory tract (13, 14). Interestingly, despite sharing a common structure throughout the respiratory tract, the cellular composition and response of the airway epithelium vary by anatomical location, with nasal and bronchial epithelial cells responding differently (16–18).

To establish colonization, *B. pertussis* employs a diverse array of virulence factors, which include adhesins such as filamentous hemagglutinin (FHA), fimbriae, and pertactin, as well as complement evasion factors and toxins such as adenylate cyclase toxin and pertussis toxin (19). These factors have been shown to disarm the innate immune responses of phagocytic cells, subvert the maturation of dendritic cells, and modulate the responses of bronchial epithelial cells (20–23). In addition to these well-characterized virulence factors, *B. pertussis*, similar to its close relative *Bordetella bronchiseptica*, encodes a type III secretion system (T3SS), which injects the cytotoxic effector protein BteA directly into host cells (24–26). While the T3SS in *B. bronchiseptica* is critical for persistent colonization in animal models, including mice, rats, and pigs (27–30), its role in *B. pertussis* pathogenesis is less clear due to poor expression or absence in laboratory-adapted strains (31, 32). However, clinical isolates of *B. pertussis* have an active T3SS that delivers a less cytotoxic variant of the BteA effector than *B. bronchiseptica* due to the insertion of A503 into BteA (33). Despite its reduced cytotoxicity, BteA of *B. pertussis* still induces cellular signaling and remodeling of the endoplasmic reticulum and mitochondrial network in HeLa cells (34).

In this study, we investigated the responses of primary human nasal epithelial cells (hNECs) cultured at an air-liquid interface (ALI) to infection with *B. pertussis*. The hNECs were isolated from the nasal airway mucosa of healthy adult donors aged 29–33 years, all of whom had received the wP vaccine in childhood. None of the donors had received a booster vaccination or had a diagnosed pertussis infection. When cultured at ALI, primary nasal and bronchial epithelial cells can differentiate into a polarized pseudostratified epithelium that closely resembles human airway epithelium (35, 36). However, although ALI epithelial cultures offer significant advantages over traditional submerged epithelial cell cultures, including cell polarity and differentiation, they still lack the full complexity of *in vivo* airway tissue. Specifically, they do not incorporate

interactions with immune and non-immune cells, resident microbiota, or biomechanical forces such as airflow, all of which influence epithelial responses in the natural airway environment. Despite the limitations of this model, polarized and differentiated hNECs represent the initial site of contact with *B. pertussis* during infection, and they have been relatively underutilized in experimental studies compared to primary bronchial epithelial cells and immortalized bronchial cell lines (23, 37). To address this gap, we infected hNECs at ALI with *B. pertussis* strain B1917, a European clinical isolate previously used in controlled human infection studies by the PERISCOPE consortium (38), and analyzed their responses. In addition, we investigated the role of the effector protein BteA in modulating the cellular responses of hNECs during infection.

## RESULTS

### Characterization of hNECs cultured at an air-liquid interface

The utility of hNECs cultured at an ALI depends on careful handling of the cultures and differentiation of the cells. To assess the suitability of our hNEC cultures for studying interactions with *B. pertussis*, we first characterized the composition of the generated hNECs.

Primary human nasal cells were isolated from nasal brushings of healthy human donors and expanded by conditional reprogramming as previously described (35). Upon expansion, cells were seeded onto Transwell membranes and air-lifted for 28–30 days to promote differentiation. Differentiation into the major airway epithelial subsets was then assessed using multi-color flow cytometry with a sequential gating strategy, according to Bonser et al. (39). This analysis included the basal cell marker CD271 (NGFR), the secretory cell marker CD66c (CEACAM6), and the Tubulin Tracker (Tub), which identifies ciliated cells by their high content of acetylated α-tubulin in the cilia. As shown in Fig. 1A, distinct epithelial subsets were identified following the gating of single cells. Next, we examined the reproducibility of differentiation and donor-to-donor variability by quantifying subset proportions across three replicates for each of the three donors. Two major cell subpopulations were distinguishable among the single cells: CD66c$^+$CD271$^-$ and CD66c$^-$CD271$^+$ subpopulations. Almost all CD66c$^-$CD271$^+$ cells showed a low Tub signal, indicating that these cells are likely basal cells (39). The CD66c$^+$CD271$^-$ cells were divided into three subpopulations: CD66c$^{High}$Tub$^{Low}$, CD66c$^{High}$Tub$^{Mid}$, and CD66c$^{Low}$Tub$^{High}$. As shown in Fig. 1B, cell-type proportions were mostly consistent among replicates but displayed high inter-donor variability, likely reflecting individual biological differences. The CD66c$^{High}$Tub$^{Low}$ and CD66c$^{High}$Tub$^{Mid}$ cells probably correspond to secretory cells, whereas the CD66c$^{Low}$Tub$^{High}$ cells are likely ciliated cells (39). In order to confirm the accuracy of this classification, we performed fluorescence-activated cell sorting of cell subsets from two donors, followed by transcriptomic profiling (Table S1). The heat map of known marker expression (Fig. 1C) validated the subset identities (14, 39) and confirmed the reliability of the flow cytometry panel as well as the differentiation process. The selected markers shown in Fig. 1C were based on reference 39 and included CD271 (NGFR), TSLP, ITGA6, and KRT5 for basal cells, MUC5B and MUC5AC for secretory cells, and CDHR3 and FOXJ1 for ciliated cells (40–43). Based on the enrichment of marker genes, it also appears that both CD66c$^{High}$Tub$^{Low}$ and CD66c$^{High}$Tub$^{Mid}$ correspond to secretory cells.

Overall, these data demonstrate that our hNEC cultures are differentiated into key epithelial subsets when cultured at ALI and provide a suitable model to analyze the interaction of *B. pertussis* with the nasal epithelial cells.

### hNECs are permissive to *B. pertussis* infection, with the bacteria localizing to both the mucus and the epithelial surface

We then sought to determine the ability of *B. pertussis* to replicate in the hNEC model. To this end, we infected cultures of hNECs on Transwell membranes apically with *B. pertussis* strain B1917 (*Bp*WT) in the ALI medium at a multiplicity of infection (MOI) of

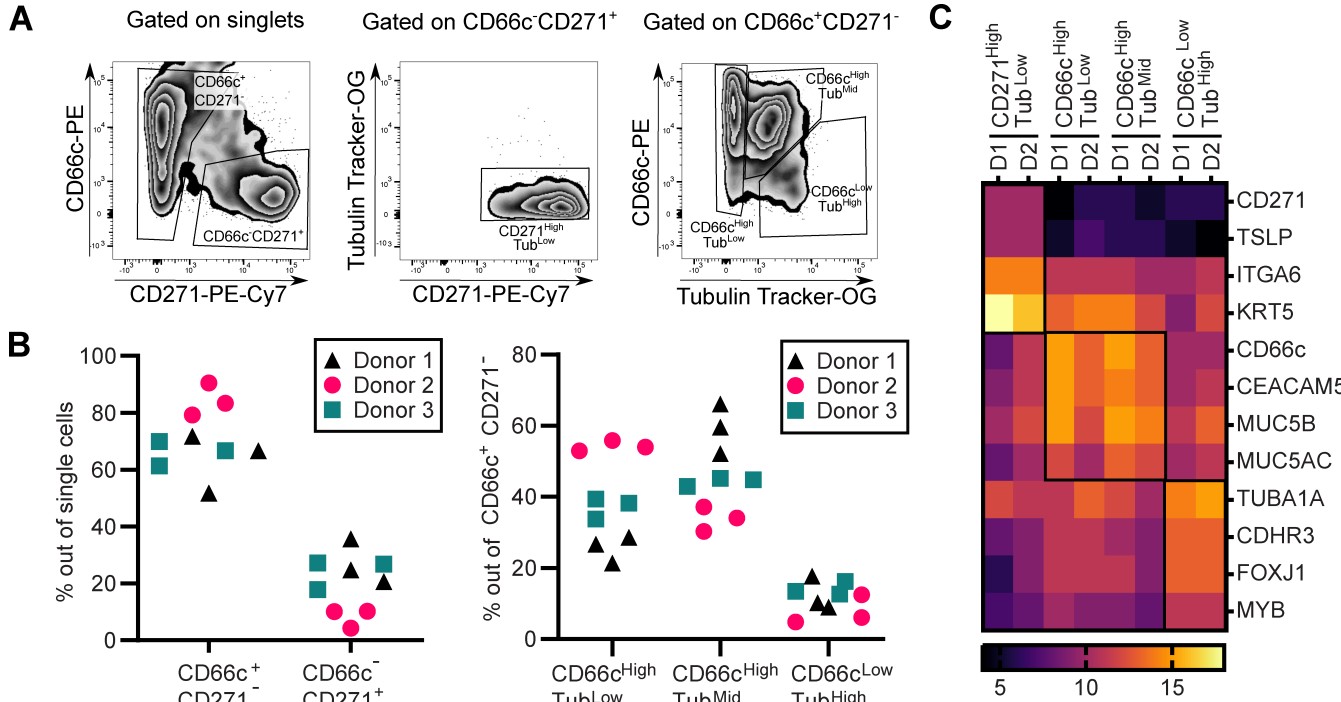

**FIG 1** Characterization of hNECs cultured at an air-liquid interface. (A) Cell gating strategy. The live single cells were gated according to staining using anti-CD271-PE-Cy7 and anti-CD66c-PE antibodies. The Tubulin Tracker-OG staining was used to refine cell subset identification. Within the CD271 subset, cells with CD271$^{High}$Tub$^{Low}$ expression were classified as basal cells. In the CD66c subset, cells with CD66c$^{High}$Tub$^{Low}$ and CD66c$^{High}$Tub$^{Mid}$ expression were classified as secretory cells, while cells with CD66c$^{Low}$Tub$^{High}$ expression were classified as ciliated cells. These gating strategies were applied to determine donor-to-donor variability and isolate cell subsets, with specific gates highlighted. (B) Donor-to-donor variability. The percentage of cell subsets was determined in three different donors using three replicate cultures of hNECs for each donor. CD66c$^{+}$CD271$^{-}$, $P < 0.01$, donor 2 vs donor 3; CD66c$^{-}$CD271$^{+}$, $P < 0.01$, donor 2 vs donor 3; CD66c$^{High}$Tub$^{low}$, $P < 0.05$, donor 1 vs donor 2; $P < 0.01$, donor 2 vs donor 3; CD66c$^{High}$Tub$^{Mid}$, $P < 0.05$, donor 1 vs donor 2; all other comparisons, non-significant; two-way ANOVA with Tukey's correction. (C) Transcriptome profiling of the cell subsets. Transcriptome profiling of the isolated cell subsets was performed in two independent experiments to evaluate transcriptional differences. Markers of basal cells: CD271 (NGFR, nerve growth factor receptor), TSLP (thymic stromal lymphopoietin), ITGA6 (CD49f, integrin subunit α6), and KRT5 (cytokeratin 5); markers of secretory cells: CD66c (CEACAM6, CEA cell adhesion molecule 6), CEACAM5 (CD66e, CEA cell-adhesion molecule 5), MUC5B (mucin 5B), and MUC5AC (mucin 5AC); markers of ciliated cells: TUBA1A (tubulin a 1A class 1), CDHR3 (cadherin-related family member 3), FOXJ1 (forkhead box J1), and MYB (MYB proto-oncogene).

10:1. After 6 hours, the medium was carefully removed, and the cultures were either processed immediately or further incubated without washing under ALI conditions. Bacterial replication was determined by the number of colony-forming units (CFUs) grown after bacterial plating at different time points post-infection. CFUs showed a 2-log increase over time, reaching $10^8$ CFUs per Transwell membrane at 72 hours (Fig. 2A). Interestingly, routine bright-field microscopy during infection revealed no visible cytotoxicity to the epithelial cell layer. Consistent with this observation and as shown in Fig. 2B, infection with an MOI of 10:1 over 24 hours did not result in significant changes in cellular composition, indicating minimal damage in the early stages of infection. These results were also confirmed by measurements of transepithelial electrical resistance (TEER), which showed no changes at 6 or 24 hours post-infection. In contrast, a significant decrease in TEER was observed at 48 and 72 hours post-infection (Fig. 2C), indicating disruption of the epithelial barrier over time. Importantly, the disruption of barrier function was independent of the T3SS effector BteA. Infections with the wild-type (WT) strain, the BteA-deficient mutant *BpΔbteA*, which has an in-frame deletion of the *bteA* gene, and the complemented strain *BpΔbteA::bteA*, which expresses BteA from the pBBRI plasmid under the control of its native promoter, all resulted in a similar decrease in TEER (Fig. 2C). As previously reported (33, 34) and shown in Fig. S1, *B. pertussis* B1917 indeed possesses an active T3SS that induces BteA-dependent cytotoxicity in HeLa

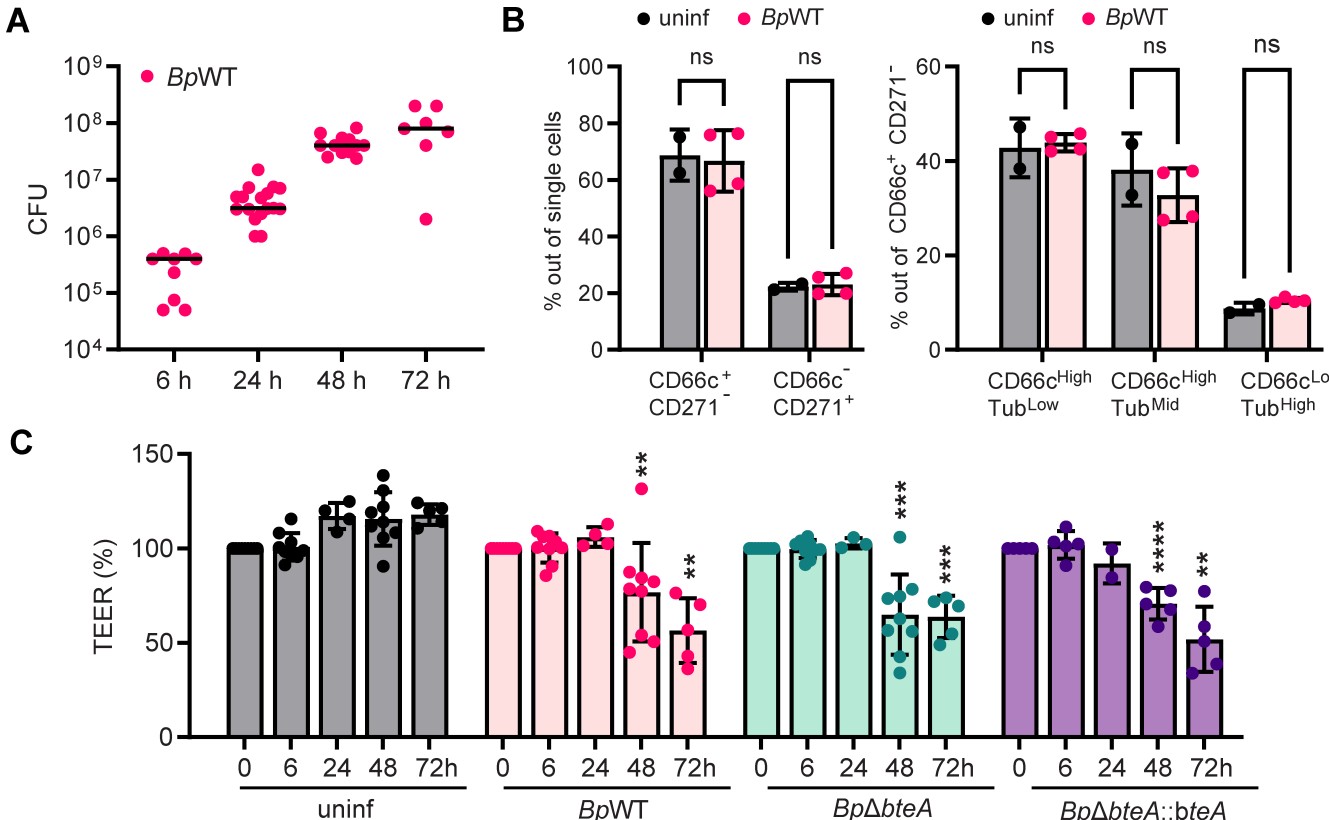

**FIG 2** hNECs support *B. pertussis* replication, leading to a gradual decline in epithelial barrier function. (A) Quantification of *B. pertussis* CFUs over time. hNECs were apically infected with *Bp*WT at a multiplicity of infection of 10:1. After 6 hours, the medium was discarded, and CFUs in the hNEC culture were quantified either immediately or after additional incubation, as indicated. Data are from two independent experiments. (B) *B. pertussis* infection does not change the composition of the hNEC culture. hNECs were apically infected with *Bp*WT at an MOI of 10:1 and analyzed by flow cytometry after 24 hours, as outlined in the legend of Fig. 1A. Non-significant (ns) at *P* < 0.05, unpaired two-tailed *t*-test. (C) TEER analysis of infected hNECs. hNECs were apically infected with *Bp*WT, *BpΔbteA*, or *BpΔbteA::bteA* at an MOI of 10:1. Transepithelial electrical resistance was measured at the indicated time points and expressed as relative TEER (in percentage), taking the starting TEER as 100%. Data are from three independent experiments. **P < 0.01, ***P < 0.001, and ****P < 0.0001 compared to uninfected controls at the corresponding time point; differences between *B. pertussis* strains were non-significant. Two-way ANOVA with Tukey's correction.

cells. The BteA-deficient strain does not induce cytotoxicity, whereas complementation partially restores it. However, during infection of hNECs, no differences in TEER decline were observed between the wild-type, mutant, and complemented strains (Fig. 2C).

To further characterize this infection model, we used fluorescence microscopy to visualize ciliated cells, *B. pertussis,* and the integrity of the epithelial barrier. As illustrated in Fig. 3A and Fig. S2, *B. pertussis* primarily did not colonize ciliated cells during the first 24 hours of infection. Localization of the bacteria to the cilia occurred sporadically at later stages of infection, being detected approximately 48 hours post-infection. Indeed, fluorescence microscopy images did not reveal any efficient recruitment of *B. pertussis* B1917 to the cell cilia. The bacteria were also found near the cell cilia but not directly attached to them. This pattern was independent of the donor and was also observed with an alternative infection protocol in which *B. pertussis* was administered in five 1 µL drops to the top of the hNEC culture on Transwell membrane (Fig. S3). Co-staining with mucin 5AC, a component of the airway mucus, further showed that *B. pertussis* localized predominantly in the mucus layer (Fig. S4). CFU analysis of phosphate-buffered saline (PBS) washes (mucus) and solubilized hNECs supported these results (Fig. 3B). The majority of bacteria were detected in the PBS washes, indicating their presence in the mucus layer, while less than 1% were firmly attached to hNECs at all time points (Fig. 3B).

Additionally, we assessed the barrier function of hNECs at different time points post-infection by examining the tight junction integrity through staining of the tight junction protein zonula occludens 1 (ZO-1). In uninfected hNECs, ZO-1 showed the expected network, which is characteristic of intact tight junctions and barrier integrity (Fig. 3A; Fig. S2). This network was not impaired at 24 hours post-infection. However, at 48 hours post-infection, we began to observe disruptions in the ZO-1 network, although some regions still retained the normal structure (Fig. 3A; Fig. S2). In contrast, at 72 hours post-infection, the ZO-1 network was largely disrupted, and the ZO-1 signal appeared relocalized, indicating a loss of tight junction integrity (Fig. 3A; Fig. S2). Interestingly, at this time point, TEER values still retained approximately 60% of their initial levels (Fig. 2C).

These results demonstrate that hNEC cultures are susceptible to *B. pertussis* infection. *B. pertussis* B1917 resides predominantly in the mucus layer and only occasionally adheres to the cilia. Importantly, the epithelial barrier remains intact during the early stages of infection and is only compromised at later time points, which is independent of the BteA effector.

## *B. pertussis* infection triggers a moderate transcriptomic response in hNECs characterized by an increase in mucin production and the absence of inflammatory signaling

To investigate how hNECs respond to infection with *B. pertussis* B1917 and assess the role of the BteA effector, we next analyzed transcriptomic changes in hNECs at 24 hours post-infection. This time point was selected as it represents an early stage of infection with minimal epithelial disruption and no detectable changes in barrier integrity or the ZO-1 network.

hNEC cultures generated from the nasal brushings of a single donor (donor 2, Fig. 1B) were apically infected with the indicated *B. pertussis* strains at an MOI of 10:1, while uninfected controls were treated with medium only. After 6 hours, the medium was carefully removed, and hNECs were incubated without washing for an additional 18 hours. Total mRNA was then isolated, followed by library preparation, sequencing, read mapping, and data analysis, as detailed in Tables S2 to S9.

As shown in Fig. 4A, principal component analysis (PCA) of the transcriptomic data revealed two main clusters. The first cluster included the data from the uninfected samples, while the second included the data from the infected samples. The transcriptomic data of the uninfected samples displayed greater variability, probably reflecting the heterogeneity of the individual Transwell membranes. In contrast, data from infected samples clustered more tightly, suggesting a shared transcriptomic response across the different bacterial strains.

Differentially expressed (DE) gene analysis identified 69 significantly modulated genes in response to *Bp*WT infection compared to uninfected controls ( $\mid$ log2FC $\mid$ $\geq$ 1; adjusted *P*-value < 0.05). The results are listed in Table S3 and illustrated in Fig. 4B, with a volcano plot showing the statistical significance (adjusted *P*-value < 0.05) against the expression change ($log_2$FC). Among these, 22 genes were upregulated, and 47 were downregulated. The upregulated genes included the mucin *MUC5AC*, the gene involved in mucin glycosylation *B3GNT6,* and the transcription factor *FOXA3*, which promotes mucus production. In contrast, genes for cytokines and chemokines, including *CSF2, IL-1α, IL-1β, IL23A,* and the family of the small proline-rich proteins (*SPRR*), possibly exhibiting bactericidal activity (44), were downregulated. These changes were observed in all infected samples, regardless of strain, comprising *Bp*WT, *BpΔbteA*, and *BpΔbteA::bteA* (Tables S4 to S6; Fig. S5A and B). Validation by RT-qPCR (Fig. 4C) confirmed significant upregulation of *B3GNT6* and *MUC5AC* genes and downregulation of the *SPRR* gene family following *Bp*WT infection. In contrast, changes in the expression of *TNF (TNF-α)*, selected based on the results of DE gene analysis, and *MUC5B,* selected for comparison with other mucus-related genes, were not statistically significant.

To assess the role of the BteA effector, we performed additional DE gene analysis comparing transcriptomic profiles of *BpΔbteA*-infected hNECs to those infected with

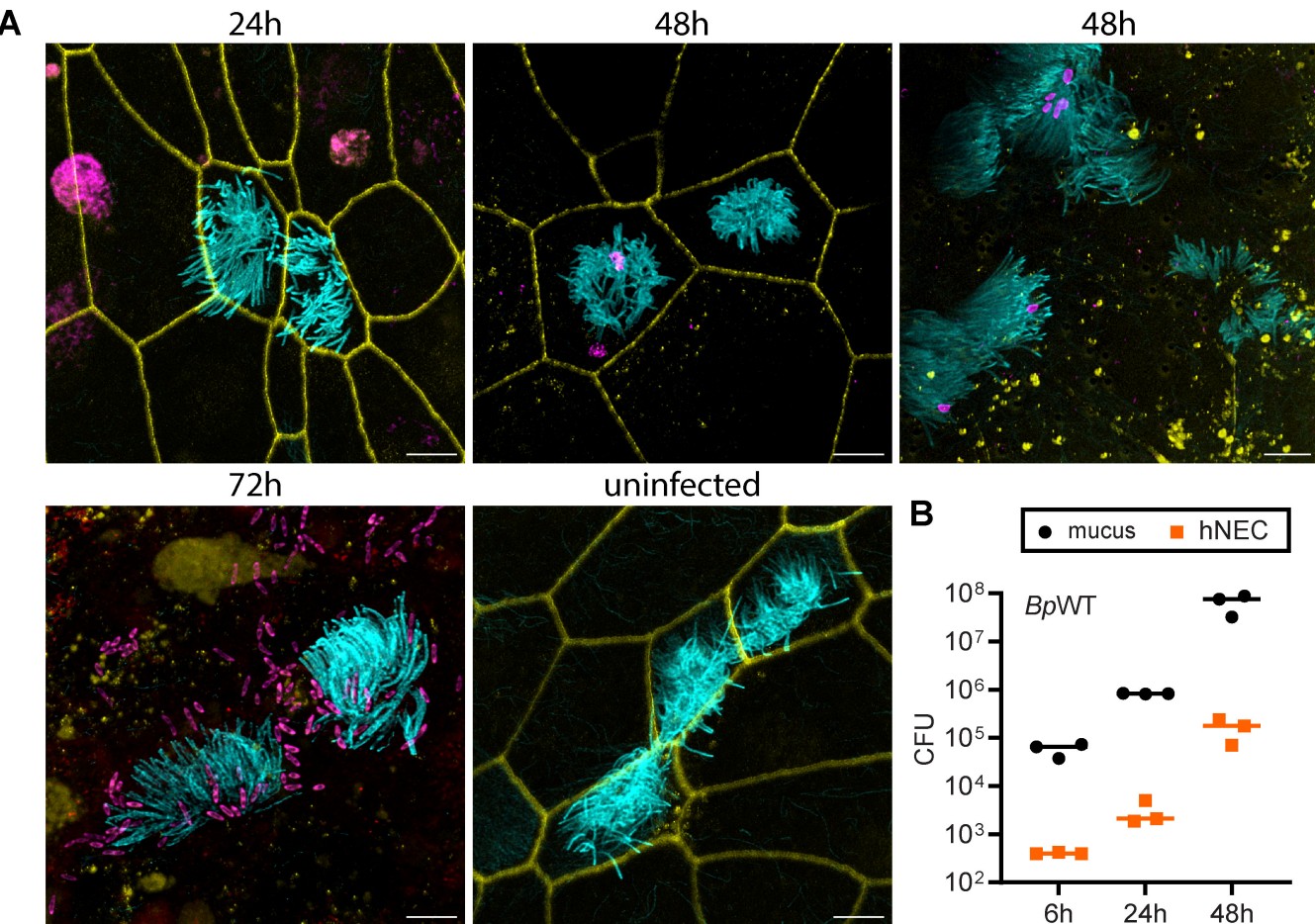

**FIG 3** *B. pertussis* localizes to both the mucus and the epithelial surface. (A) Visualization of *B. pertussis* on hNECs over time. hNECs were apically infected with *Bp*WT expressing the fluorescent mScarlet protein at an MOI of 10:1. After 6 hours, the apical medium was removed, and *Bp*WT was visualized by mScarlet expression (magenta) either immediately or after additional incubation, as indicated. The tight junctions (zonula occludens 1 [ZO-1], yellow) were stained with an anti-ZO-1 antibody, followed by an anti-rabbit IgG-DyLight-405 conjugate, while the cilia (cyan) were labeled with an anti-acetylated tubulin antibody, followed by an anti-mouse IgG-AF488 conjugate. Images represent maximum intensity (Z-max) projections from confocal Z-stack images and are representative of two independent experiments. Scale bar, 5 µm. (B) Quantification of the association of *B. pertussis* with hNECs. Cells were apically infected with *Bp*WT at an MOI of 10:1. After 6 hours, the apical medium was removed, and CFUs from the PBS washes (mucus) and cell-associated (hNEC) fractions were quantified either immediately or after additional incubation, as indicated. The data represent three independent biological replicates.

*Bp*WT and the complemented strain *BpΔbteA::bteA* (Tables S7 to S9; Fig. S5C and D). Only nine hNEC genes reached statistical significance in the *BpΔbteA* vs *Bp*WT comparison ( │ log2FC │ ≥ 1; adjusted *P*-value < 0.05). Of these, six genes were significantly downregulated, whereas three genes were significantly upregulated. The results are in accordance with PCA, where deletion of the effector BteA had a minimal impact on the overall host transcriptomic response (Fig. 4A), suggesting that BteA does not play a critical role in modulating the host transcriptome within this time interval and this infection model. Furthermore, only one of these genes, CREB-regulated transcription coactivator 2 (CRTC2), which was downregulated following *BpΔbteA* infection, showed restored expression upon infection with a complemented strain *BpΔbteA::bteA*. It is important to note, however, that complementation of the *BpΔbteA* strain only partially restored cytotoxicity against HeLa cells, likely due to a reduced efficiency of plasmid-borne BteA export (Fig. S1), which may limit interpretation of the complementation results.

Taken together, *B. pertussis* B1917 infection elicits a rather limited transcriptomic response in hNECs, characterized by the upregulation of genes involved in mucin

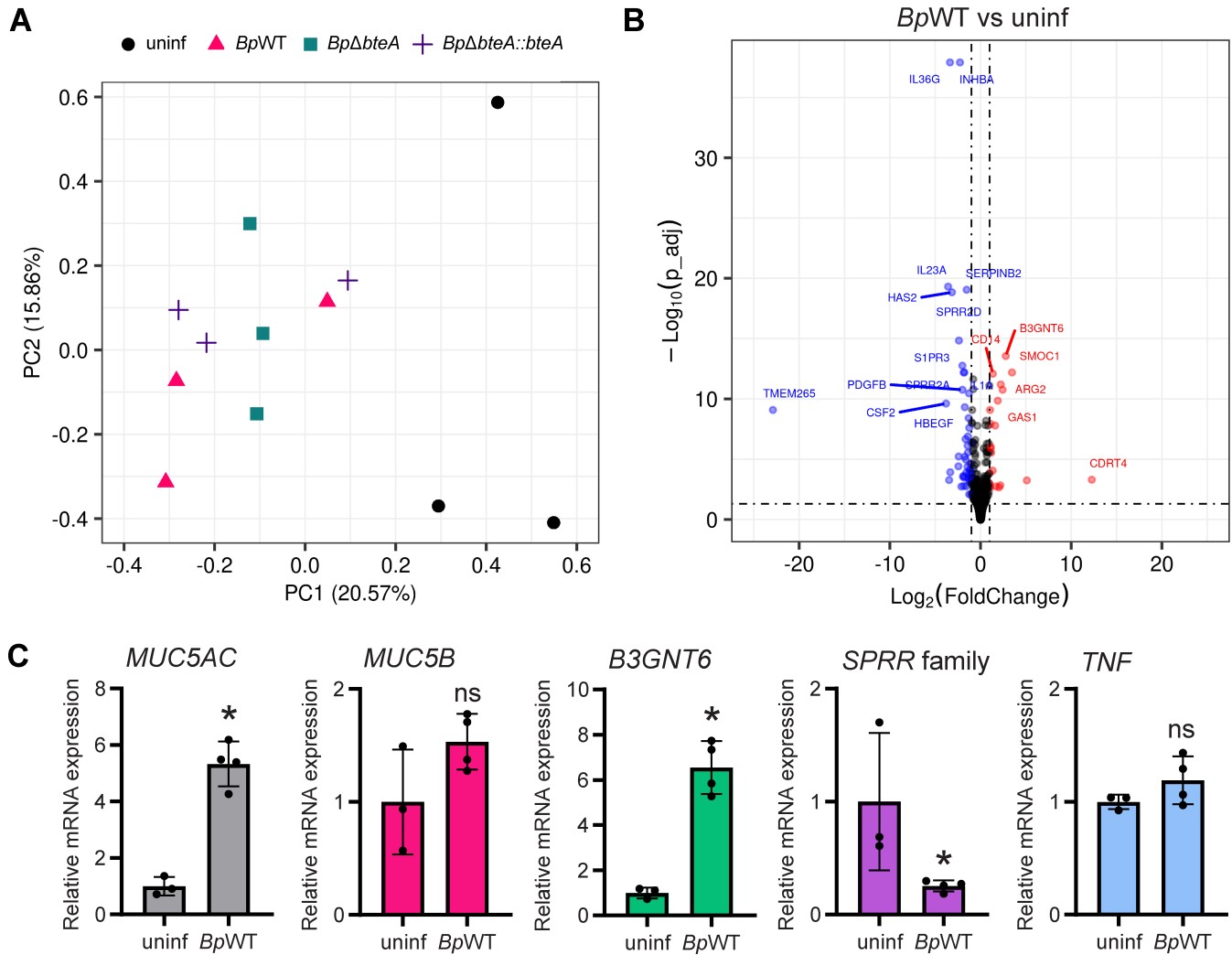

**FIG 4** *B. pertussis* induces a moderate transcriptomic response in hNECs characterized by an increase in mucin production and minimal inflammatory signaling. (A) Principal component analysis of transcriptomic data. PCA was performed on transcriptomic data from uninfected hNECs and hNECs infected with *Bp*WT, *BpΔbteA*, and *BpΔbteA::bteA* at an MOI of 10:1 for 24 hours. Each dot represents an independent biological replicate. (B) Volcano plot of differential gene expression in *Bp*WT-infected hNECs vs uninfected hNECs. Significant changes were defined as |log2 fold change| ≥ 1 and adjusted *P*-value ≤ 0.05. Red dots represent significantly upregulated genes, and blue dots represent significantly downregulated genes. Black dots indicate genes with nonsignificant changes. (C) Validation of RNA-seq results by quantitative qRT-PCR. Relative mRNA expression levels of MUC5AC, MUC5B, B3GNT6, SPRR family, and TNF (TNF-α) were analyzed using qRT-PCR in *Bp*WT-infected hNECs vs uninfected controls. Samples were collected 24 hours post-infection, and relative expression changes were quantified. Each dot represents an independent hNEC culture.

production and the absence of chemokine and inflammatory cytokine induction. The BteA effector does not have a significant impact on transcriptomic response in the context of this infection model.

## Proteomic changes in hNECs confirm increased mucin production

To further characterize the *B. pertussis*-induced response in hNECs, we performed proteomic analysis at 48 hours post-infection. This later time point was selected to allow sufficient time for the infection to progress and for the transcriptomic changes to be reflected at the protein level. At this time point, the barrier function of the hNEC was clearly impaired (Fig. 2C and 3A).

hNEC cultures, derived from the nasal brushings of a single donor (donor 3, Fig. 1B), were apically infected with the indicated strains at an MOI of 10:1, as described above.

After 6 hours, the medium was carefully removed, and hNECs were maintained under ALI conditions for an additional 42 hours. The proteins were extracted, processed for analysis, and subjected to LC-MS/MS. The acquired data were searched against both *B. pertussis* and *Homo sapiens* databases and analyzed as detailed in Tables S10 to S18.

Analysis of *B. pertussis* proteins, listed in Table S10, revealed the presence of critical adhesins, including FHA, fimbriae (Fim3), and pertactin, as well as the autotransporters BrkA and Vag8. Pertussis toxin and adenylate cyclase toxin were also detected, although their presence varied across individual hNEC cultures. This variability likely reflects the average bacterial load of $5 \times 10^7$ per Transwell (see Fig. 2A), which approached the detection threshold in our experimental setup due to the abundance of hNEC proteins. No significant differences in the expression of *B. pertussis* proteins were observed between experimental groups. Some components of the T3SS, including the translocator protein BopD, were detected in some of the infected samples; however, the effector BteA was not detected (Table S10).

PCA of the human proteome profiles (excluding *B. pertussis* proteins, Table S11), shown in Fig. 5A, revealed the clustering of replicates within their respective groups. Interestingly, there was no clear separation between uninfected and *Bp*WT-infected samples, whereas the profiles of samples infected with the complemented strain *BpΔbteA::bteA* showed the clearest separation. This unexpected divergence suggests that the complemented strain may not fully replicate the wild-type infection.

DE analysis identified 20 significantly abundant proteins in *Bp*WT-infected samples compared to uninfected controls ( | log2FC |  ≥ 1; adjusted *P*-value < 0.05), as documented in Table S12 and visualized by the volcano plot in Fig. 5B. Among these, two proteins were upregulated, and 18 were downregulated. Importantly, mucin MUC5AC was significantly upregulated following *BpWT* infection (Fig. 5B). Upregulation of mucins MUC5AC and MUC5B was also observed in samples infected with *BpΔbteA* and *BpΔbteA::bteA* strains (Tables S13 to S15; Fig. S6A and B). In contrast, proteins involved in extracellular matrix remodeling, matrix metalloproteinases 1 and 9 (MMP1, MMP9),

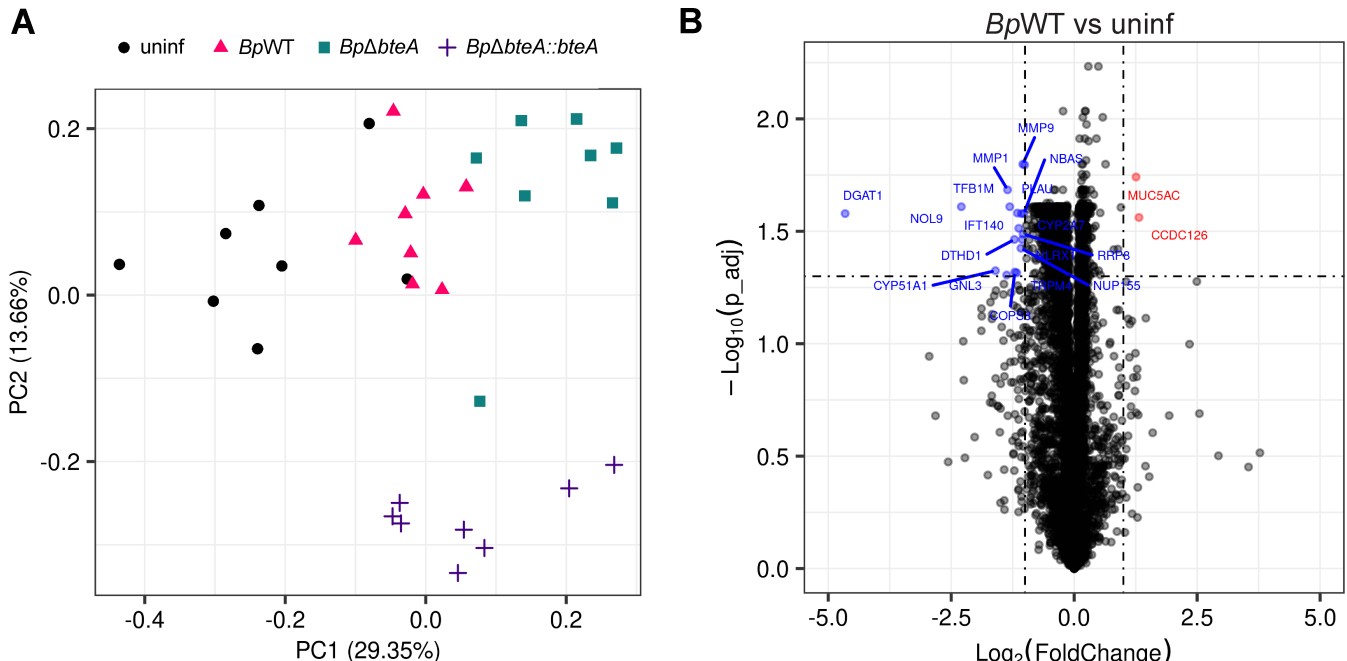

**FIG 5** Proteomic changes in hNECs confirm increased mucin production. (A) Principal component analysis of proteomic data. PCA was performed on proteomic data from uninfected hNECs and hNECs infected with *Bp*WT, *BpΔbteA,* and *BpΔbteA::bteA* at an MOI of 10:1 for 48 hours. Each dot represents an independent biological replicate. (B) Volcano plot of differential protein expression in *Bp*WT-infected hNECs vs uninfected hNECs. Significant changes were defined as |log2 fold change| ≥ 1 and adjusted *P*-value ≤ 0.05. Red dots represent significantly upregulated proteins, and blue dots represent significantly downregulated proteins. Black dots indicate proteins with nonsignificant changes.

and plasminogen activator urokinase were significantly downregulated in all infected samples (Table S15).

To assess the role of BteA, we next compared proteomic profiles of hNECs infected with *BpΔbteA* to those infected with *Bp*WT and the complemented strain *BpΔbteA::bteA*. The comparison of *BpΔbteA* vs *Bp*WT (Table S16; Fig. S6C) identified 26 significantly modulated host proteins, of which 5 were upregulated and 21 were downregulated. Among these, only two proteins, tyrosine-protein kinase (BAZ1B) and pleckstrin homology-like domain family B member 2 (PHLDB2), showed restored expression in the *BpΔbteA::bteA*-infected samples (Tables S17 and S18; Fig. S6D). However, the interpretation of these results may be limited due to the unexpected divergence in the PCA profile of the complemented strain, indicating possible inconsistencies in its behavior compared to the wild-type infection.

Taken together, nevertheless, these results are consistent with the transcriptomic data and confirm that *B. pertussis* B1917 infection enhances mucin production in hNECs. The overall proteomic response is moderate and exhibits variability. Proteomic analysis revealed that the T3SS effector BteA does not play a major role in modulating the host proteome in this infection model.

## DISCUSSION

This work investigates the interaction of the clinical isolate *B. pertussis* B1917 with primary human nasal epithelial cells cultured at ALI, which are the first cells encountered by *B. pertussis* during respiratory tract infection. We characterized the kinetics of *B. pertussis* B1917 interaction with hNECs, their early host transcriptomic and proteomic responses, and the role of the T3SS effector BteA in modulating these responses. We provide a comprehensive transcriptomic and proteomic data set for this model. Despite donor-to-donor variability, which is inherent to primary cell models, our analyses revealed increased mucin production and limited inflammatory signaling using both approaches.

In this model, *B. pertussis* B1917 is primarily localized to the mucus layer with minimal attachment to the cell cilia up to 24 hours post-infection. This observation contrasts with previous studies in rabbit tracheal explants and primary human bronchial cells, where *B. bronchiseptica* RB50, *B. pertussis* Tohama I, and its streptomycin-resistant derivative BP536 rapidly associated with epithelial cilia within minutes to 6 hours post-infection (45–47). Several factors may underlie these differences, including the amount of mucus in the experimental system, the ability of *B. pertussis* B1917 to penetrate the mucus layer, or potential differences in cilia properties and mucus composition between the models. In addition, allelic variations or differential expression of adhesins, including fimbriae (Fim2 or Fim3), filamentous hemagglutinin, and pertactin in *B. pertussis* B1917 compared to other strains could contribute to the reduced attachment (45, 46, 48, 49). Nevertheless, bronchial and nasal mucus composition has been reported to be highly similar, with only subtle differences (50). Moreover, our proteomic analysis (Table S10) confirmed the expression of Fim3, FHA, and pertactin by *B. pertussis* B1917. However, changes in their expression levels or contributions of different alleles compared to the above-mentioned *Bordetella* strains cannot be excluded.

To further address this discrepancy, we also compared ciliary recruitment of *B. pertussis* B1917 and Tohama I (generously provided by Dr. Holubova, Institute of Microbiology, Czech Republic) in our model system. By applying five 1 µL drops directly to the apical surface of hNECs cultured on a Transwell membrane, we were able to observe B1917 on the cell surface or in close proximity at 24 hours post-infection, with only sporadic presence in the cell cilia. In contrast, Tohama I was abundantly present in the cilia at the same time point (Fig. 6; Fig. S7). The underlying reasons for the reduced ciliary recruitment of *B. pertussis* B1917 compared to Tohama I are unclear, and it will be important to test the behavior of other currently circulating *B. pertussis* strains in future studies.

## *Bp*WT B1917 *Bp*WT Tohamal

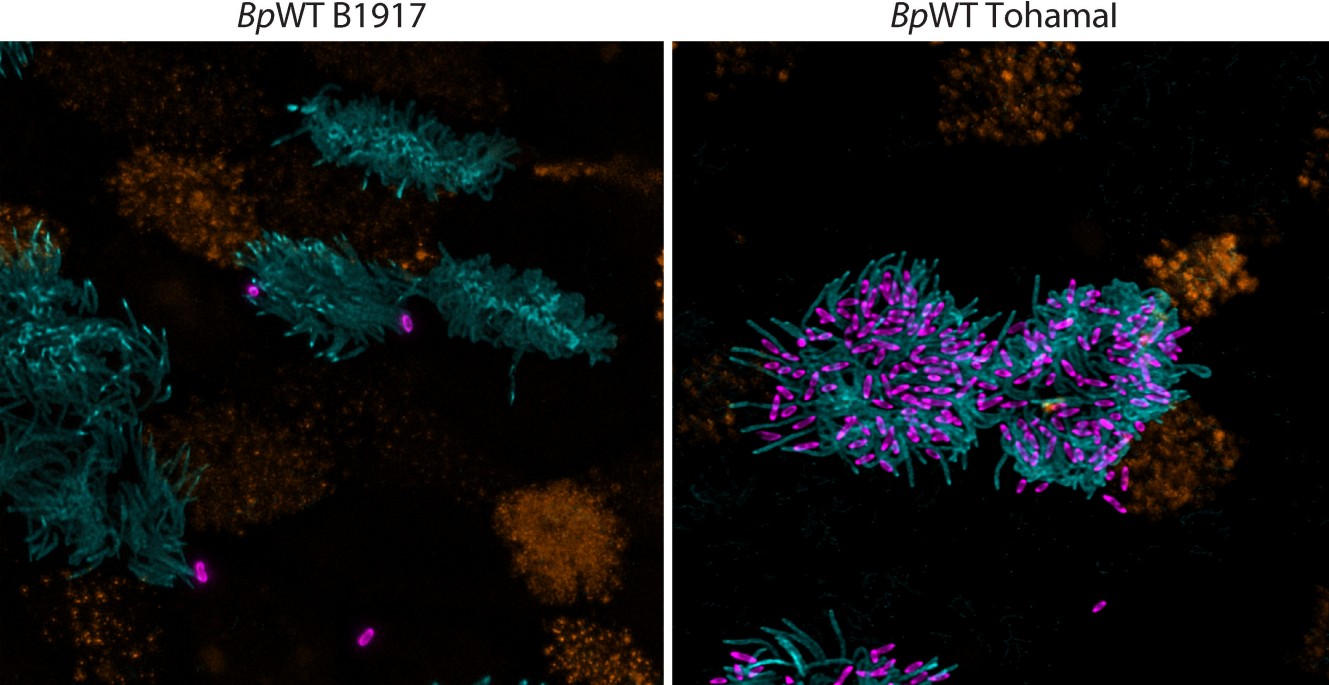

**FIG 6** Visualization of *B. pertussis* B1917 and *B. pertussis* Tohama I on hNEC. *Bp*WT strain B1917, used throughout the study, or *B. pertussis* wild-type Tohama I strain expressing fluorescent mScarlet protein were deposited on the top of the hNECs on Transwell membrane using five 1 µL drops at an MOI of 10:1. After 24 hours, the B1917 or Tohama I bacterial cells were visualized by mScarlet expression (magenta). The cilia (cyan) were stained with an anti-acetylated tubulin antibody followed by an anti-rabbit IgG-AF488 conjugate, while mucin 5AC inside secretory cells (orange) was labeled with an anti-MUC5AC antibody followed by an anti-mouse IgG-DyLight-405 conjugate. Images represent maximum intensity (Z-max) projections from confocal Z-stack images and are representative of two independent experiments. Scale bar, 5 µm.

The colonization of the mucus layer by *B. pertussis* B1917 is consistent with previous findings from *in vivo* studies, which report high *B. pertussis* loads in nasopharyngeal washes from infected baboons (51). Interestingly, the major protein components of mucus are mucin glycoproteins that mainly terminate with sialic acid or fucose (52, 53). Both *B. pertussis* and *B. bronchiseptica* have been shown to possess sialic acid-binding capacity, which enables them to bind mucin (54–56). In addition, mucin can inhibit the attachment of *B. pertussis* to A549 lung carcinoma cells (54), suggesting that mucus may simultaneously function as a niche for bacterial persistence and as a barrier to prevent epithelial attachment. Such localization of *B. pertussis* in the mucus layer may protect epithelial cells from structural damage and limit their activation. Although *B. pertussis* can invade and survive in respiratory epithelial cells (57–59), the invasion rate in primary nasal and bronchial cells is exceptionally low (60), consistent with its extracellular pathogenesis.

In our model, epithelial integrity was maintained up to 24 hours post-infection. A more pronounced disruption of hNEC barrier function was detected only at 48 hours post-infection. This disruption was independent of the T3SS effector BteA, despite its demonstrated cytotoxicity in HeLa cells (33, 34) and Fig. S1. A likely factor responsible for the barrier disruption is the adenylate cyclase toxin (ACT), which has been previously shown to impair epithelial integrity through cAMP-mediated signaling (23, 59). Nevertheless, additional studies are needed to clarify the specific contribution of ACT in this model.

To further investigate the responses of hNECs to *B. pertussis* infection and assess the role of BteA, we next performed both transcriptomic and proteomic analyses. The transcriptomic analysis was performed at 24 hours post-infection, a time point characterized by minimal epithelial disruption and limited pathology. At this time point, tight

junctions of epithelial cells remained intact. As such, we would not expect transcriptomic changes originating from epithelial damage and impairment of the tight junction network to be prominent. Proteomic analysis was performed 48 hours post-infection, allowing sufficient time for transcriptomic changes to be reflected at the protein level. At this later time point, *B. pertussis* had undergone replication, and the barrier function of the hNEC layer was compromised.

The potential impact of donor vaccination on transcriptomic and proteomic responses is unclear. All hNEC donors in this study had received wP vaccination in childhood, but none had received a booster dose or had a history of confirmed pertussis infection. Whether hNECs from unvaccinated individuals or those vaccinated with aP vaccine would respond differently is unknown. Primary nasal epithelial cells were collected by nasal brushing and differentiated under ALI conditions for 28–30 days. Successful differentiation was confirmed by flow cytometry using cell-type-specific markers. Individual cultures, however, exhibited considerable heterogeneity, with different proportions of cell types observed both within and between donors. This variability was more pronounced among the different donors (Fig. 1B) but was also reflected in the transcriptome and proteome data sets for individual donors, as shown in the PCA plots (Fig. 4A and 5A).

Despite this heterogeneity, transcriptomic and proteomic analyses revealed a consistent but moderate response to *B. pertussis* infection. At 24 hours post-infection, 69 mRNA transcripts were significantly modulated, while 20 proteins showed significant changes at 48 hours post-infection. Importantly, we observed robust upregulation of mucin-related genes, particularly the mucin MUC5AC, at both transcript and protein levels. MUC5AC and MUC5B are the major secreted gel-forming mucins of the airways (61). Increased mucus production could support bacterial clearance by the mucociliary escalator. However, while MUC5B is critical for mucociliary clearance, MUC5AC is not essential for this function. Instead, tethering of MUC5AC to the epithelium has been associated with mucociliary dysfunction and mucus plugging in asthma (61–63). In addition, *B. pertussis* has been shown to impair mucociliary clearance by inducing ciliostasis through tracheal cytotoxin, a muramyl tetrapeptide derived from the bacterial cell wall (37, 64, 65). This impairment can lead to the stagnation of mucus and the formation of a mucus gel matrix, which could not only impair clearance but also create a niche for colonization.

The observed upregulation of MUC5AC is consistent with previous studies, where cAMP signaling mediated by ACT induced mucin production in both the bronchial epithelial cell line VA10 and the mouse model (23, 66). However, MUC5AC expression can also be driven by various other stimuli, including Toll-like receptor (TLR) activation followed by NF-κB signaling or through elevated levels of inflammatory cytokines (67, 68). Interestingly, in our model, both TLR activation and inflammatory cytokine production appeared to be limited. This contrasts with the strong inflammatory responses observed in infected macrophages or bronchial epithelial cell lines (56, 69). One possible explanation is that TLRs and/or their cofactors, such as MD-2, which are essential for effective TLR signaling, may be expressed at lower levels or are restricted to the basolateral surface, which limits their activation (70, 71). This spatial restriction is typically lost in epithelial cells cultured as submerged monolayers or when barrier integrity is compromised during infection, which potentially accounts for the observed discrepancy between the model systems. Indeed, previous studies using ALI cultures of bronchial epithelial cells infected with even higher MOI of *B. pertussis* also reported only modest immune activation (37).

Importantly, in an infected host, epithelial cell responses are likely to be modulated by a variety of soluble factors, interactions with immune and non-immune cells, and also the surrounding microbial community. For example, IL-1β and IFNγ present in conditioned media of *B. pertussis*-infected macrophages and NK cells have been reported to be essential for robust chemokine secretion by bronchial epithelial cells in response to *B. pertussis* infection (37, 72). In addition, TLR expression on epithelial cells is known

to be upregulated during inflammation or in the presence of the microbial community (70). Consistent with this, our transcriptomic data showed modest upregulation of CD14, the lipopolysaccharide (LPS) transferase that supports the binding of LPS to a complex of TLR4-MD2 (73) following *B. pertussis* infection. However, it is important to mention that *B. pertussis* does not produce conventional LPS. Instead, it expresses lipooligosaccharide with a truncated, non-repeating O-chain and only five acyl chains on lipid A, which might affect its stimulatory properties (74, 75).

Epithelial cells also contribute to mucosal defense through the production of antimicrobial peptides, including members of the defensin family and most likely also the small proline-rich proteins SPPR (44, 76). In our model system, we observed a slight downregulation of SPRR transcripts, while expression of β-defensins remained unchanged. Previous studies have demonstrated that the T3SS of *B. bronchiseptica* suppresses β-defensin expression through inhibition of NF-κB signaling pathway (77, 78). In contrast, in our study, the T3SS effector BteA of *B. pertussis* exhibited only a limited effect in hNECs, as it neither impaired epithelial integrity nor did it substantially alter hNEC responses to infection, including antimicrobial peptide expression. These results suggest that BteA activity may be highly context-dependent and possibly restricted to specific host cell types or conditions.

The efficiency of BteA translocation is likely influenced by complex regulatory mechanisms that control T3SS activity. Regulation occurs at multiple levels, including transcriptional, post-transcriptional, and even post-translational control through sigma factor BtrS (BrpL), anti-sigma factor BtrA (BspR) (79–81), the partner-switcher proteins BtrU, BtrV, and BtrW encoded in the *btr* locus (82, 83), and T3SS gatekeeper protein BopN (84). Furthermore, global regulators, including the RNA chaperone Hfq and the secondary messenger c-di-GMP, have also been implicated in modulating T3SS activity in *Bordetella* species (85–88). This regulatory network likely responds to environmental signals that remain poorly defined. Indeed, T3SS genes are upregulated in *B. pertussis* following mouse infection compared to *in vitro* culture conditions (89, 90). In our current study, proteomic analysis at 48 hours post-infection failed to detect BteA and most of the structural components of the injectisome. This absence may reflect detection sensitivity or, alternatively, a downregulation of T3SS expression. Nevertheless, it is important to emphasize that control experiments verified that the used *B. pertussis* strains expressed a functional T3SS at the time of infection, as shown in Fig. S1.

A plausible explanation for the limited activity of the T3SS in our model system is the localization of *B. pertussis* B1917 within the mucus layer. Given that T3SS-dependent effector delivery requires contact with the host cell membrane, the physical separation may impair injectisome function. This contrasts with the action of secreted virulence factors, such as ACT and pertussis toxin, which can still intoxicate epithelial cells even in the absence of direct bacterial contact.

Interestingly, infection of hNECs with the *BpΔbteA* mutant still induced several significant changes in host gene expression compared to cells infected with the wild-type strain, comprising 9 transcripts and 26 proteins. However, only three of these changes were restored upon complementation with BteA. These included (i) CRTC2, which was downregulated at the transcript level; (ii) tyrosine-protein kinase BAZ1B; and (iii) pleckstrin homology-like domain family B member 2 (PHLDB2), both decreased at the protein level. Although these targets were regulated in a BteA-dependent manner, they do not appear to be directly linked to the previously described BteA-mediated disruption of calcium homeostasis and organelle fragmentation in HeLa cells (34). Nevertheless, their regulatory potential is intriguing. CRTC2, originally named TORC2, acts as a co-activator of CREB and responds to both Ca$^{2+}$ and cAMP signaling (91). BAZ1B is a chromatin remodeling factor (92), while PHLDB2, also known as LL5β, contributes to cortical cytoskeleton organization and has been implicated in the modulation of EGFR signaling (93, 94). It should be emphasized, however, that the complementation of the *BpΔbteA* strain did not fully restore BteA-dependent cytotoxicity in HeLa cells (Fig. S1). Moreover, PCA of the proteomic data revealed an unexpected divergence of

the complemented strain from both the wild type and mutant, warranting cautious interpretation of the complementation data. Among the host proteins whose changes remained dysregulated despite BteA reintroduction are additional proteins implicated in gene regulation, chromatin structure, and cell metabolism (Tables S9 and S18). Further studies will be essential to validate these candidate targets and elucidate their potential role. Equally important will be the identification of host-derived signals that control the expression of BteA and the activity of the T3SS injectisome.

In conclusion, this study provides a detailed analysis of the interaction between *B. pertussis* B1917 and differentiated hNECs. The infection was characterized by delayed disruption of epithelial barrier integrity, limited immune activation, enhanced mucin production, and localization of bacteria within the mucus layer. All this may allow *B. pertussis* to evade early host defense mechanisms and establish a productive infection during the early stages of respiratory tract colonization.

## MATERIALS AND METHODS

### Bacterial strains, growth conditions, and plasmid complementation

The bacterial strains and plasmids used in this study are listed in Table S19. The *B. pertussis* B1917 WT strain was kindly provided by Dr. Branislav Vecerek (Institute of Microbiology, Czech Academy of Sciences). It was received in May 2014 from Prof. Frits R. Mooi (National Institute of Public Health and the Environment, Bilthoven, The Netherlands) through Dr. Marjolaine van Gent. The *B. pertussis* B1917 in-frame gene deletion mutants were previously generated using the suicide allelic exchange vector pSS4245 (33). For complementation of the *B. pertussis* B1917 Δ*bteA* mutant, we used the pBBRI plasmid construct described in the same study, which encodes the *bteA* allele of *B. pertussis* B1917 under its native promoter (33). For the visualization of *B. pertussis* B1917 during infection, *B. pertussis* B1917 harboring the pBBRI plasmid with the *B. bronchiseptica* RB50 GroES promoter (PgroES) and coding sequence of the mScarlet protein (mSc) was used. *B. pertussis* Tohama I (Institute Pasteur collection #CIP 81.32) harboring pBBRI vector with *B. pertussis* Tohama I filamentous hemagglutinin promoter (PfhaB) and coding sequence of the mScarlet protein (mSc) (22) was a kind gift from Dr. Jana Holubova (Institute of Microbiology, Czech Academy of Sciences).

*Escherichia coli* strains were cultivated at 37°C in Luria-Bertani (LB) agar or LB broth, with chloramphenicol (15 µg/mL) added to the medium when appropriate. *B. pertussis* strains were grown on Bordet-Gengou (BG) agar medium (Difco, USA) supplemented with 1% glycerol and 15% defibrinated sheep blood (Lab-MediaServis, Jaromer, Czech Republic) at 37°C and 5% $CO_2$. For experiments, *B. pertussis* was grown at 37°C to the mid-exponential phase ($OD_{600}$ 1.0) in a modified Stainer-Scholte (SSM) medium with reduced L-glutamate (monosodium salt) concentration (11.5 mM, 2.14 g/L) and without $FeSO_4 \cdot 7H_2O$ to maximize T3SS expression (33), as specified in Table S20.

For the introduction of the pBBRI plasmid into *B. pertussis* B1917, bacterial conjugation with *Escherichia coli* strain SM10λ pir was utilized. *B. pertussis* strains harboring pBBRI plasmid were selected on BG agar supplemented with 15 µg/mL of chloramphenicol and 100 µg/mL of cephalexin to which *B. pertussis* B1917 is naturally resistant. Colonies were then re-streaked onto BG agar with chloramphenicol (15 µg/mL) to maintain the pBBRI plasmid.

### BteA analysis in bacterial whole-cell lysates and culture supernatants

*B. pertussis* strains were cultivated in the SSM medium at 37°C. For whole-cell lysate analysis, culture aliquots were centrifuged (30 min; 15,000 *g*), and pellets were lysed in 8 M urea and 50 mM Tris-HCl, pH 8.0. Lysates were then mixed with SDS-PAGE sample loading buffer. To analyze supernatant fractions, cell-free culture supernatants were precipitated overnight at 4°C with 10% trichloroacetic acid, washed with acetone, resuspended in 8 M urea, 50 mM Tris-HCl, pH 8.0, and mixed with SDS-PAGE sample

loading buffer. Samples corresponding to $OD_{600}$ equivalent to 0.1 (whole-cell lysates) or 1 (bacterial supernatants) were separated by SDS-PAGE electrophoresis using a 10% gel and transferred onto the nitrocellulose membrane. Membranes were incubated overnight with anti-BteA mouse serum (dilution 1:10,000), followed by horseradish peroxidase-conjugated anti-mouse IgG secondary antibodies (1:3,000 dilution, GE Healthcare). Detection was performed using a Pierce ECL chemiluminescence substrate (Thermo Fisher Scientific, USA) and an Image Quant LAS 4000 system (GE Healthcare, USA).

## Culture of cell lines

The feeder cell line for human nasal epithelial cells, 3T3-J2 (Kerafast, Cat. #EF3003, embryonic mouse fibroblasts), was cultured in DMEM with 10% bovine calf serum and antibiotics (0.1 mg/mL streptomycin and 1,000 U/mL penicillin), as specified in Table S21. To arrest the cell proliferation of feeder 3T3-J2 cells, mitomycin C was added to the medium at a final concentration of 4 µg/mL, and cells were incubated for 2 hours at 37°C and 5% $CO_2$.

HeLa cell line (ATCC, Cat. #CCL-2, human cervical adenocarcinoma) was cultured in Dulbecco's Modified Eagle Medium (DMEM) with 10% heat-inactivated fetal bovine serum (FBS, DMEM-10%FBS, Table S21) at 37°C and 5% $CO_2$.

## Air-liquid interface culture of human nasal epithelial cells

Human nasal epithelial cells were collected from the nasal brushings of healthy donors using a cytology brush. The collected cells were digested with TrypLE Select (Gibco, Cat. #12604013) for 5 min at 37°C to yield a single cell suspension. The cells were then centrifuged (5 min; 350 $g$) and gently resuspended in NEC medium, specified in Table S21.

To allow for their conditional reprogramming and expansion, hNECs were seeded into a T25 flask on mitomycin-treated 3T3-J2 fibroblasts in the NEC medium, as previously reported (35). Once the hNEC population had sufficiently expanded, $5 \times 10^4$ cells were plated onto the apical side of a collagen-coated 6.5 mm Transwell membrane (Corning Costar, Cat. #3470, pore size 0.4 µm) in 200 µL of apical and 600 µL of basolateral NEC medium without fungin. After 72 h, during which the cells reached confluency, the apical medium was removed (air-lifting), and the basolateral medium was replaced by the differentiation ALI medium (Table S21). The ALI medium was replaced three times per week. The air-lifting defined day 0 of the ALI culture, and the experiments were conducted between days 28 and 30. To prevent excessive mucus accumulation on the apical side, the apical surfaces of cells were washed with phosphate-buffered saline (PBS) for 30 min every 7 days, starting from day 14 to 17 of the ALI culture and depending on the mucus buildup as assessed by visual inspection.

The differentiation and epithelial barrier integrity of ALI cultures were monitored through visual inspection and transepithelial electrical resistance measurements before each experiment. Additionally, on day 25 of the ALI culture, the basolateral medium was replaced with an antibiotic-free ALI medium, with a minimum of two medium changes before experiments.

## Multi-color flow cytometry of hNECs

Multi-color flow cytometry was used to characterize differentiated hNECs. Prior to processing for flow cytometry, hNECs cultured on Transwell membranes were stained with 2× concentrated Tubulin Tracker Green (Oregon Green 488 Taxol, bis-acetate, Invitrogen, Cat. #T34078), applied apically in 200 µL of HBSS. The cells were incubated for 1 hour at 37°C in a 5% $CO_2$ atmosphere. After incubation, the solution was discarded, and the hNECs were washed twice apically with 200 µL of HBSS. To detach the hNECs from the Transwell membranes, the cells were incubated for 30 min in Accutase solution (Invitrogen, Cat. #00-4555-56) supplemented with Collagenase D (1 mg/mL, Roche, Cat.

#11088858001) and DNase I (200 μg/mL, Roche, Cat. #10104159001). For this process, 600 μL of the solution was added to the bottom compartment, and 200 μL was applied apically. Following the incubation, the released cells were collected, centrifuged at 350 $g$ for 5 min, washed once with flow cytometry buffer (5% fetal calf serum and 2 mM EDTA in PBS), and resuspended in flow cytometry buffer containing anti-CD66c (clone B6.2, Exbio, Cat. #. 1P-863-T100) and anti-CD271 (clone NGFR5, Exbio, Cat. #. T7-642-T100) (Table S22). Antibody staining was performed on ice for 1 hour. Positive staining was determined using fluorescence-minus-one controls to ensure accurate gating and analysis. Data were acquired using a BD LSRII flow cytometer and analyzed using FlowJo version 10 software. Statistical analysis was done in GraphPad Prism (version 10.3.0) using two-way ANOVA with Tukey's correction or unpaired two-tailed $t$-test, as appropriate.

## RNA sequencing of flow cytometry-sorted populations

The cells were stained for multi-color flow cytometry of hNECs as described above, which was followed by sorting of cells on a BD Influx cell sorter. Sorted cells were collected directly into the RLT lysis buffer, with at least $2 \times 10^4$ cells collected per population. Total RNA was extracted according to the instructions of the RNeasy Plus Micro Kit (Qiagen, Cat. #74134). RNA concentration and quality were assessed using the DS-11 spectrophotometer DeNovix. In addition, RNA integrity was analyzed using an Agilent Bioanalyzer 2100 (Agilent Technologies, USA) (see Fig. S8A). Only samples with a minimum RNA integrity number (RIN) of 7 were included for library preparation.

The libraries were prepared using the SMARTer Stranded Total RNA-Seq Kit version 2 (Takara Bio, Cat. #634411), using 2 ng of RNA as input per sample. Sequencing was performed on the Illumina NextSeq 500 platform with 75 bp single-end reads. A total of 60 million reads and 30 million reads per sample were obtained for experiments 1 and 2, respectively. Raw sequencing reads were preprocessed with the nf-core/RNAseq pipeline (version 3.10.1) using the default settings for STAR + Salmon (95, 96). Reads were mapped to the ENSEMBL reference genome and transcriptome (*Homo_sapiens* GRCh38.dna_sm.primary assembly genome and GRCh38.110 transcriptome). Gene counts were normalized in RStudio (version 4.4.2) using the package DESeq2 (version 1.44.0) (97) and then transformed with regularized logarithm (rlog). Subsequently, the batch effect was removed by the package limma (98). Visualization of the heat map was done by GraphPad Prism (version 10.3.0).

The data have been deposited with links to BioProject accession number PRJNA1216212 in the NCBI BioProject database (https://www.ncbi.nlm.nih.gov/bioproject/). GEO accesion number is GSE302040.

## Infection of hNECs and determination of bacterial colony-forming units

Differentiated ALI cultures of hNECs were infected apically with $3 \times 10^6$ CFU of *B. pertussis* B1917 in 200 μL of ALI medium. After 6 hours, the medium was carefully removed without washing. The cultures were either processed immediately or further incubated under ALI conditions for a total infection time of 24, 48, and 72 hours.

To determine total bacterial CFUs, 200 μL of Accutase solution (Invitrogen, Cat. #00-4555-56) was added to the apical compartment, and 600 μL of the same solution was added to the bottom compartment after discarding the basal medium. The cultures were incubated for 30 min at 37°C and 5% $CO_2$ to facilitate cell detachment. After incubation, the apical suspension was carefully pipetted up and down to release the detached cells and transferred to a tube containing 200 μL PBS supplemented with 2% TX-100. Given that the Transwell membrane had a pore size of 0.4 μm, which prevents bacterial passage to the bottom compartment, the Accutase solution from the basal side was subsequently re-used to wash the apical compartment and collect the remaining material. The collected material was pooled with the initial apical suspension.

To obtain mucus-associated bacteria, 200 μL of PBS was added to the apical compartment, and the culture was incubated for 15 min. The apical wash was then

collected, and the process was repeated once more. The pooled mucus-containing suspension was serially diluted in PBS and plated on BG agar plates. After the removal of the mucus-associated bacteria, the bacteria remaining on the Transwell membrane were considered to be firmly attached. To determine their number (CFUs), the cultures were processed as described for total bacterial CFUs. Bacterial colonies were counted after 5 days.

## Determination of barrier function by measurements of trans-epithelial electrical resistance

To monitor the barrier function of hNECs during differentiation, trans-epithelial electrical resistance was measured using a Millicell-ERS volt-ohm meter (Millipore, USA). For assessing the barrier function of hNECs following infection with indicated *B. pertussis* strains, TEER measurements were performed using the ECIS Z Theta system (Applied Biophysics) with a plate mounted on the ECIS station. At the indicated time points, 200 µL of PBS was added to the apical side of hNEC cells, and TEER values were recorded. The background resistance of an empty Transwell membrane was subtracted from all measurements. Changes in TEER were calculated relative to the TEER value measured at time point 0 for each individual Transwell membrane. Statistical analysis was done using two-way ANOVA with Tukey's correction in GraphPad Prism (version 10.3.0).

## Cytotoxicity assay against HeLa cells

The cytotoxicity of *B. pertussis* B1917 and its mutant derivatives toward HeLa cells was determined by monitoring changes in cell membrane integrity using the fluorescent DNA binding dye CellTox Green (Promega, Cat. #G8743), as described previously (99). In brief, $2 \times 10^4$ of HeLa cells per well were seeded in a 96-well black/clear bottom plate (Corning, USA) in DMEM-10%FBS. The next day, *B. pertussis* strains were added at an MOI of 50:1 together with CellTox Green reagent. The plate was centrifuged (5 min; 120 $g$) and placed inside the chamber at 37°C and 5% $CO_2$ of the TecanSpark microplate reader (Tecan, Switzerland). Fluorescence measurements at $494_{ex}/516_{em}$ with a 10 nm bandwidth for both were performed at defined time intervals for a 16-hour period.

## Immunofluorescent staining and image acquisition

Differentiated ALI cultures of hNECs grown on Transwell membranes were infected apically with $3 \times 10^6$ CFU of the *B. pertussis* B1917 expressing the monomeric red fluorescent protein mScarlet under the control of the constitutive *gro*ES promoter (*Bp*WT/mSc, see Table S19). mScarlet exhibits exceptional brightness and quantum yield within its spectral class and allows high-quality fluorescence imaging (100). It represents a highly stable protein and does not serve as an indicator of bacterial metabolic activity or viability.

Infections were carried out using 200 µL of ALI medium, which was carefully removed without washing after 6 hours of incubation, or using 5 µL of SS medium administered as 1 µL drops. At the indicated time points, Transwell membranes were fixed with prewarmed 4% PFA (Santa Cruz) for 20 min at RT, followed by washes with PBS (3× for 5 min). Furthermore, the hNECs on Transwell membranes were permeabilized with 0.2% Triton-X100 (TX-100) in PBS for 10 min at RT. The Transwell membranes were then blocked with 4% BSA in PBS supplemented with 0.05% TX-100 (PBST) for 1 hour at RT. For immunostaining, the following primary antibodies were applied overnight at 4°C in 1% BSA-PBS, as required: anti-ZO-1 rabbit antibody (ThermoFisher, Cat. #339100), anti-acetylated tubulin mouse antibody (Sigma/Merck, Cat. #T6793), anti-acetylated tubulin rabbit antibody (ThermoFisher, Cat. #MA5-33079), and anti-MUC5AC mouse antibody (ThermoFisher, Cat. #MA5-12178). See Table S22 for working dilutions. The next day, Transwell membranes were washed with PBST (3× for 5 min) and incubated for 1 hour at RT with appropriate secondary antibodies in 1% BSA-PBST: anti-rabbit IgG-DyLight-405 conjugate (Jackson Immunoresearch, Cat. #111-475-003),

anti-mouse IgG-AF488 conjugate (Jackson Immunoresearch, Cat. #115-546-062), anti-rabbit IgG-AF488 conjugate (Jackson Immunoresearch, Cat. #111-546-144), or anti-mouse IgG-DyLight-405 conjugate (Jackson Immunoresearch, Cat. #715-476-150), as required. After incubation, Transwell membranes were washed with PBST (3× for 5 min) and briefly rinsed in dH$_2$O. The membranes were then dissected from Transwells, placed inside AD-Seal mounting spacers (ADVI, Cat. #ADS-12-07100), attached to glass slides, and covered with the AD-Mount-F mounting medium (ADVI, Cat. #ADM-001). Finally, the membranes were sealed with coverslips No. 1.5H (Marienfeld, Cat. # 0107032).

Confocal images were acquired with Leica STELLARIS 8 equipped with a wide-range (440–790 nm) light laser with the pulse picker (WLL PP) and highly sensitive hybrid detectors operated by the LAS X software. The objective HC PL APO 40×/1.25 GLYC CORR CS2, WD 0.35 mm, was used with the type G immersion (Leica, Cat. #11513859). DyLight-405 was excited with a 405 nm DMOD laser, and other dyes with the WLL-PLL laser, with the following settings: AF488 (491 nm) and mScarlet (561 nm). Z-stacks were acquired with steps of 0.15 µm, and pixel size was set to meet Nyquist sampling criteria. Lightning Expert in the LasX software was used for the deconvolution. Deconvolved images were inspected and processed (brightness/contrast adjustments) with the ImageJ/FIJI software (101), and final figures were prepared in Adobe Illustrator.

## Transcriptomic analysis of bulk RNAseq and statistical analysis

At 24 hours post-infection, hNECs on the Transwell membranes were lysed using 0.3 mL of RLT lysis solution (Qiagen RNeasy plus micro kit, Cat. #74134) per membrane. Homogenization was carried out by repeated pipetting to ensure complete cell disruption. Total RNA was then isolated according to the instructions of the RNeasy plus micro kit (Qiagen, Cat. #74134) using a single Transwell membrane per sample with the DNase I treatment. During the RW1 wash step, 80 µL of the RDD buffer containing 10 µL of DNase I solution was pipetted directly onto the spin column and incubated for 15 min at RT. RNA concentration and quality were assessed using the DS-11 spectrophotometer DeNovix, and RNA integrity was analyzed using an Agilent Bioanalyzer 2100 (Agilent Technologies, USA) (see Fig. S8B). Samples with a minimum RIN of 6 were used in library preparation.

The libraries were prepared using the KAPA mRNA HyperPrep Kit with polyA selection (Roche, Cat. #08098123702), with 300 ng of RNA as input per sample. Sequencing was performed on the NextSeq 2000 Illumina platform with 122 bp single-end reads. At least 25 million reads per sample were obtained. Raw sequencing reads were preprocessed with the nf-core/RNAseq pipeline (version 3.10.1) using the default settings for STAR + Salmon (95, 96). Reads were mapped to the ENSEMBL reference genome and transcriptome (Homo_sapiens GRCh38.dna_sm.primary assembly genome and GRCh38.110 transcriptome). Principal component analysis and differential expression gene analysis were performed using the package DEseq2 (version 1.44.0) (97) in RStudio (version 4.4.2). For both analyses, only genes containing at least 10 reads across all samples were included. PCA was plotted using ggplot2 (version 3.5.1) after data transformation with regularized logarithm (rlog package DEseq2). Adaptive shrinkage (ashr [102]) was applied to log2 fold changes to stabilize effect size estimates before DE analysis visualization using ggplot2 (version 3.5.1). Only genes with a |log2 fold change| ≥ 1 and adjusted $P$-value ≤ 0.05 were considered significantly differentially expressed.

The data have been deposited with links to BioProject accession number PRJNA1214286 in the NCBI BioProject database (https://www.ncbi.nlm.nih.gov/bioproject/). GEO accession number is GSE288608.

## Quantitative real-time PCR

RNA isolated as described in the transcriptomic analysis was reverse transcribed into cDNA using the High-Capacity cDNA Reverse Transcription Kit (ThermoFisher Scientific, Cat. #4368814). qPCR was performed using EvaGreen reagent (Solid Biodyne, Cat. #08-25-00001) with 20 ng of cDNA used per reaction, with data collected using a CFX384

PCR instrument (BioRad). Two housekeeping genes, *RPL13A* and *GAPDH*, were used as reference genes. The sequences of the qPCR primers and their efficiencies are provided in Table S23 and Fig. S9. Data analysis was done using Biorad CFX Maestro software (Biorad, USA).

## Protein extraction and sample preparation for proteomics

At 48 hours post-infection, hNECs on Transwell membranes were lysed with 0.15 mL of preheated (90°C) sodium deoxycholate (SDC) lysis buffer containing 2% SDC and 100 mM Tris-HCl, pH 7.4, per membrane. Lysis was carried out for 20 s, with complete cell homogenization achieved by repeated pipetting. The lysates were transferred to tubes and heated at 95°C for 5 min. To maximize protein recovery, the membranes were washed with an additional 0.15 mL of preheated SDC buffer, and the wash was pooled with the initial extraction. Each sample corresponded to a single Transwell membrane.

Lysates were boiled at 95°C for 10 min in 100 mM triethylammonium bicarbonate (TEAB) containing 2% SDC, 40 mM chloroacetamide, and 10 mM tris(2-carboxyethyl)phosphine and further sonicated (Bandelin Sonoplus Mini 20, MS 1.5). Protein concentration was determined using the BCA protein assay kit (ThermoFisher), and 10 µg of protein per sample was used for MS sample preparation. The sample volume was then adjusted to 65 µL in total by adding 100 mM TEAB containing 2% SDC. Samples were further processed using SP3 beads on the Thermo KingFisher Flex automated Extraction & Purification System in a 96-well plate. Briefly, 65 µL of the sample was added to 65 µL of 100% ethanol and mixed with the SP3 beads. After binding, the beads were washed three times with 80% ethanol. After washing, samples were digested in 50 mM TEAB at 40°C with 1 µg of trypsin for 2 hours, then another 1 µg of trypsin was added and digested overnight. After digestion, samples were acidified with TFA to 1% final concentration, and peptides were desalted using in-house-made stage tips packed with C18 disks (Empore) according to reference 103.

## LC-MS/MS and data analysis

LC separation was performed on the Dionex Ultimate 3000 nano HPLC system online connected to the MS instrument. Samples were loaded onto the trap column (C18 PepMap100, 5 µm particle size, 300 µm × 5 mm, Thermo Scientific) for 2.000 min at 17.500 µL/min. The loading buffer was composed of water, 2% acetonitrile, and 0.1% trifluoroacetic acid. Peptides were eluted with a mobile phase B gradient from 4.0% to 35.0% B in 64.0 min. Mobile phase buffer A was composed of water and 0.1% formic acid. Mobile phase B was composed of acetonitrile and 0.1% formic acid. A nano reversed-phase column (Aurora Ultimate TS, 25 cm × 75 µm ID, 1.7 µm particle size, Ion Opticks) was used for LC/MS analysis.

The peptide mixture was analyzed on Thermo Scientific Orbitrap Ascend using a data-independent approach. Eluting peptide cations were converted to gas-phase ions by electrospray ionization in positive mode. The spray voltage was set to 1,600 V, and the ion transfer tube temperature was set to 275°C. MS1 scans of peptide precursors were analyzed in Orbitrap in the range of 145–1,450 *m/z* at a resolution of 60,000 and with the following settings: RF Lens 60%, maximum injection time 123 ms, and AGC target 100. DIA scans were performed in Orbitrap at a resolution of 30,000. The AGC target was set to 1,000, and the maximum injection time mode was set to Auto. Precursor mass range 400–1,000 *m/z* was covered by 30 windows, each 20 Da wide. Activation type was set to HCD with 28% collision energy.

All data were analyzed and quantified with the Spectronaut 19 software (104) using directDIA analysis. Data were searched against the Human database (downloaded from Uniprot.org in March 2024 containing 20,598 entries) and *Bordetella pertussis* (downloaded from Uniprot.org in June 2024 containing 3,258 entries). Enzyme specificity was set as C-terminal to Arg and Lys, also allowing cleavage at proline bonds and a maximum of two missed cleavages. Carbamidomethylation of cysteines was set as a fixed modification, and N-terminal protein acetylation and methionine oxidation as variable

modifications. FDR was set to 1% for PSM, peptide, and protein. Quantification was performed on MS2 level. Precursor PEP cut-off and precursor and protein cut-off were set to 0.01, and protein PEP was set to 0.05. Label-free quantification intensity data were exported (Table S10), and their analysis was performed using Perseus (version 1.6.15.0) (105).

To analyze differences in the human proteome after *B. pertussis* infection, PCA and DE analyses were employed. Only proteins detected in at least six of the eight biological replicates per experimental group were included in the analysis. The missing values were imputed using sampling from a normal distribution. Statistical significance was determined by non-paired Welch's *t*-test, and *P*-values were adjusted using the Benjamini and Hochberg method. Only proteins with a |log2 fold change| ≥ 1 and adjusted *P*-value ≤ 0.05 were considered significantly differentially expressed. All data visualizations were done via RStudio (version 4.4.2) with package ggplot2 (version 3.5.1). The mass spectrometry proteomics data have been deposited to the ProteomeXchange Consortium via the PRIDE (106) partner repository with the data set identifier PXD060316.

## ACKNOWLEDGMENTS

We thank Sarka Knoblochova, MD, from Institute of Microbiology, Czech Academy of Sciences, for assistance with nasal brushings of healthy donors, Michal Kolar, PhD and Miluse Hradilova, PhD from Genomics and Bioinformatics Core Facility, Institute of Molecular Genetics, Czech Academy of Sciences for valuable advices and bioinformatic help, and Karel Harant, PhD and Pavel Talacko, PhD from the Mass Spectrometry and Proteomics Service Laboratory, Faculty of Science, Charles University for performing the LC-MS/MS runs. We would also like to acknowledge the support of the project LM2023053 (Czech National Node to the European Infrastructure for Translational Medicine) from the Ministry of Education, Youth and Sports of the Czech Republic.

This work was supported by the grant 21-05466S of the Czech Science Foundation (www.gacr.cz), grant Talking microbes- understanding microbial interactions within One Health framework (CZ.02.01.01/00/22_008/0004597) of the Ministry of Education, Youth and Sports of the Czech Republic (www.msmt.cz) and the Lumina Queruntur Fellowship LQ200202001 of the Czech Academy of Sciences to J.K.

## AUTHOR AFFILIATION

[1]Laboratory of Infection Biology, Institute of Microbiology of the Czech Academy of Sciences, Prague, Czech Republic

## AUTHOR ORCIDs

Martin Zmuda http://orcid.org/0000-0002-1672-4420
Ivana Malcova http://orcid.org/0000-0002-4385-3842
Jana Kamanova http://orcid.org/0000-0001-6574-4404

## FUNDING

| Funder | Grant(s) | Author(s) |
| --- | --- | --- |
| Grantová Agentura České Republiky | 21-05466S | Jana Kamanova |
| Ministerstvo Školství, Mládeže a Tělovýchovy | CZ.02.01.01/00/22_008/0004597 | Jana Kamanova |
| Akademie Věd České Republiky | LQ200202001 | Jana Kamanova |

## AUTHOR CONTRIBUTIONS

Martin Zmuda, Data curation, Formal analysis, Investigation, Methodology, software, Validation, Visualization, Writing – original draft | Ivana Malcova, Data curation, Formal analysis, Investigation, Methodology, Validation, Visualization | Barbora Pravdova,

Investigation | Ondrej Cerny, Formal analysis, Investigation | Denisa Vondrova, Investigation | Jana Kamanova, Conceptualization, Data curation, Formal analysis, Funding acquisition, Project administration, Resources, Supervision, Validation, Visualization, Writing – original draft

## DATA AVAILABILITY

The RNA sequencing data from flow cytometry sorted populations are available in the NCBI Sequence Read Archive (SRA) under BioProject number PRJNA1216212, GEO accession number GSE302040. Bulk RNA sequencing data are accessible under BioProject number PRJNA1214286, GEO accession number GSE288608. Additionally, the mass spectrometry proteomics data have been deposited to the ProteomeXchange Consortium with the dataset identifier PXD060316.

## ETHICS APPROVAL

The sampling of the nasal cavity was conducted on healthy donors who provided written informed consent for the use of their cells in research. The study received approval under EK-297/24 of the Ethics Committee of Motol University Hospital, Prague, Czech Republic.

## ADDITIONAL FILES

The following material is available online.

### Supplemental Material

**Supplemental figures (Spectrum01267-25-s0001.pdf).** Fig. S1 to S9.
**Table S1 (Spectrum01267-25-s0002.xlsx).** TPM values of gene expression in cell subsets from donor 1 and donor 2.
**Table S2 (Spectrum01267-25-s0003.xlsx).** TPM values of all gene expressions at 24 hours post-infection.
**Supplemental tables (Spectrum01267-25-s0004.xlsx).** Tables S3 to S9. Gene expression analysis.
**Supplemental tables 2 (Spectrum01267-25-s0005.xlsx).** Tables S10 and S11. LFQ proteomic data.
**Supplemental tables 3 (Spectrum01267-25-s0006.xlsx).** Tables S12 to S18. Proteomic analysis.
**Supplemental tables 4 (Spectrum01267-25-s0007.pdf).** Tables S19 to S23. Description of bacterial strains and plasmids, composition of cultivation media, and used antibodies and qRCR primers.

### Open Peer Review

**PEER REVIEW HISTORY (review-history.pdf).** An accounting of the reviewer comments and feedback.

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
