## [Reviewer comments · Microbiology Spectrum]

Microbiology Spectrum

Limited response of primary nasal epithelial cells to *Bordetella pertussis* infection

Martin Zmuda, Ivana Malcova, Barbora Pravdova, Ondrej Cerny, Denisa Vondrova, and Jana Kamanova

Corresponding Author(s): Jana Kamanova, Mikrobiologicky ustav Akademie ved Ceske republiky

Review Timeline:

Submission Date:	April 24, 2025
Editorial Decision:	May 19, 2025
Revision Received:	June 20, 2025
Accepted:	June 26, 2025

Editor: Catherine Brissette

Reviewer(s): The reviewers have opted to remain anonymous.

Transaction Report:

DOI: <https://doi.org/10.1128/spectrum.01267-25>

Re: Spectrum01267-25 (**Limited response of primary nasal epithelial cells to *Bordetella pertussis* infection and the effector protein BteA**)

Dear Dr. Jana Kamanova:

Thank you for the privilege of reviewing your work. Below you will find my comments, instructions from the Spectrum editorial office, and the reviewer comments.

Please address the comments by both reviewers. I don't believe additional experiments are warranted, but adjust conclusions accordingly and provide any necessary clarifications.

Revision Guidelines

Sincerely,
Catherine Brissette
Editor
Microbiology Spectrum

Reviewer #1 (Comments for the Author):

In this study, the authors use a nasal epithelial model to investigate the interaction of *Bordetella pertussis* with what would be the initial site of infection in the host. They examined the response of this cell type to infection and the potential role of the Type III Secretion System (TTSS) in epithelial colonization. The manuscript is well-written and easy to read. It has the merit of being one

of the few studies that model the initial interaction of Bp with its host. There are, however, certain details that require attention.

Regarding the CFU counts, it is not clear to this reviewer what the authors aim to assess with the experimental design. The authors should clarify what information they intend to obtain by washing the apical compartment with the Accutase solution collected from the basal compartment, after the cells have been detached using Accutase. It would be helpful if the authors could explain the rationale behind this sampling approach.

Additionally, it is surprising that the authors do not analyze the fractions separately. For example, quantifying CFUs in the lower compartment could help assess and potentially confirm the presumed disruption of tight junctions. By the way, what is the pore size of the Transwell filters?"

Regarding the association of bacteria with the mucus, it is worth noting that, although microscopy shows the presence of bacteria in the mucus and previous studies indicate that *B. pertussis* adheres to mucin, the presence of bacteria in the mucus does not necessarily imply that they are replicating there. While CFU counts in PBS samples suggest the presence of viable bacteria on the apical side of the cells, this is a very indirect piece of evidence, as the bacteria could originate from the well or other areas outside the mucus.

To support the claim that mucus is a site of bacterial replication, the authors need to provide direct evidence. There are numerous tools available to test this hypothesis, should the authors choose to pursue this line of investigation. If not, the conclusions should be revised accordingly.

Finally, it would be important to demonstrate that the bacteria attached to the mucus are alive. Is mScarlet a vital stain with a half-life suitable for assessing viability in this context? The authors should include this information in the manuscript.

Is there any evidence that the TTSS is expressed and active in BP1917 under the conditions of the assay? If not, and considering the ongoing controversy surrounding the TTSS in *B. pertussis*, it might be more appropriate to omit the experiments with the mutant strains. Furthermore, since the results obtained with the complemented strain in Figure 5 cannot be explained, this control should not be considered valid in any experiment.

The information presented in Figure 6S regarding the different adhesion sites of the reference strain Tohama I and the clinical isolate B1914 is highly significant, as it appears to suggest that modes of adhesion to nasal epithelial cells may have evolved over time. This finding could have important implications for understanding the adaptations this pathogen has undergone during its continued circulation in the population. It may represent one of the important results of this study and should be included in the main body of the manuscript.

Reviewer #2 (Comments for the Author):

This paper by Zmuda et al describe the responses of *Bordetella pertussis* to nasal epithelial cells in a primary cell culture. Pertussis disease is still causing a significant amount of deaths in infant under 5 years of age. Importantly, subclinical infections with *B. pertussis* allowing for constant circulation within the population. The authors develop a primary nasal epithelial cell culture model, that allows for the investigation of first responses from host as well as first adaptations of the bacterium to the host environment. This is a well written manuscript that present the development of a new in vitro model to investigate first responses of human nasal epithelia to *Bordetella pertussis* as well as the adaptation of *Bordetella pertussis* to human nasal epithelia. This manuscript validates the model which presents itself very useful, while the authors also state the limitations that need to be consider when using the model. Some comments are included for the authors:

1. The authors discuss that ACT mutant might contribute (Line 370) to barrier disruption. Adding a deletion mutant and complemented strain to confirm this hypothesis should be included in the barrier disruption experiments.
2. CRTC2 downregulated in the BteA mutant and restored in the complemented strain. CRTC2 can act as regulator of Ca signaling. The authors have previously described that BteA disrupt calcium homeostasis. A discussion is needed as this might be relevant to further understand the role of BteA as well as can serve as foundation for future investigations.
3. BAZ1B and PHLDB2 are also restored upon BteA complementation in the proteomic analysis while they are also not discussed. BAZ1B discussion might not be relevant at this point due to its reported function up to date. However, the role of PHLDB2 is to regulate EGFR which is an epidermal growth factor, and it can be relevant in the context of the model employ in these experiments.
4. Finally, more references are needed specially in discussion and methods to be able to go to original sources

Response to the reviewers of the manuscript Spectrum01267-25, entitled: "Limited response of primary nasal epithelial cells to *Bordetella pertussis* infection and the effector protein BteA" by Zmuda M. *et al.*, now renamed to "Limited response of primary nasal epithelial cells to *Bordetella pertussis* infection" to better reflect the overall scope of the study.

We thank both reviewers for their insightful suggestions, which helped us to improve our manuscript. Below we provide a point-by-point response to each comment.

Reviewer #1:

General comment:

In this study, the authors use a nasal epithelial model to investigate the interaction of Bordetella pertussis with what would be the initial site of infection in the host. They examined the response of this cell type to infection and the potential role of the Type III Secretion System (TTSS) in epithelial colonization. The manuscript is well-written and easy to read. It has the merit of being one of the few studies that model the initial interaction of Bp with its host. There are, however, certain details that require attention.

Comments:

1. Regarding the CFU counts, it is not clear to this reviewer what the authors aim to assess with the experimental design. The authors should clarify what information they intend to obtain by washing the apical compartment with the Accutase solution collected from the basal compartment, after the cells have been detached using Accutase. It would be helpful if the authors could explain the rationale behind this sampling approach.

Additionally, it is surprising that the authors do not analyze the fractions separately. For example, quantifying CFUs in the lower compartment could help assess and potentially confirm the presumed disruption of tight junctions. By the way, what is the pore size of the Transwell filters?"

A1: We thank the reviewer for pointing this out. We have now clarified in the manuscript that the Transwell inserts used had a pore size of 0.4 μm . This pore size allows hNEC culture but prevents bacterial passage to the bottom compartment, even after disruption of the epithelial barrier at 48 hours post-infection. It is also important to note that 99% of bacteria at all time points were present in the apical washes.

In the setup for determination of the firmly attached bacteria or total bacteria, the medium from the bottom compartment was first discarded, followed by addition of Accutase to digest the hNEC layer from the basal side. After 30 minutes, the Accutase was re-used to help collect the remaining material from the apical side. The reason was simply to economize the use of Accutase solution. We apologize for the confusion and have revised the method section of the manuscript to clarify this. See highlighted lines 641-644.

2. Regarding the association of bacteria with the mucus, it is worth noting that, although microscopy shows the presence of bacteria in the mucus and previous studies indicate that B. pertussis adheres to mucin, the presence of bacteria in the mucus does not necessarily imply that they are replicating there. While CFU counts in PBS samples suggest the presence of

viable bacteria on the apical side of the cells, this is a very indirect piece of evidence, as the bacteria could originate from the well or other areas outside the mucus.

To support the claim that mucus is a site of bacterial replication, the authors need to provide direct evidence. There are numerous tools available to test this hypothesis, should the authors choose to pursue this line of investigation. If not, the conclusions should be revised accordingly.

Finally, it would be important to demonstrate that the bacteria attached to the mucus are alive. Is mScarlet a vital stain with a half-life suitable for assessing viability in this context? The authors should include this information in the manuscript.

A2: In response, we have carefully revised wording in the manuscript and avoid referring to the bacteria in the mucus, as replicating bacteria. We simply state their localization in the mucus layer. We have also included additional information about mScarlet, clarifying that it is a stable fluorescent protein expressed under the control of the constitutive GroES promoter (highlighted lines 680-684). It is not a viability marker and does not indicate metabolic activity.

*3. Is there any evidence that the TTSS is expressed and active in BP1917 under the conditions of the assay? If not, and considering the ongoing controversy surrounding the TTSS in *B. pertussis*, it might be more appropriate to omit the experiments with the mutant strains. Furthermore, since the results obtained with the complemented strain in Figure 5 cannot be explained, this control should not be considered valid in any experiment.*

A3: To address these concerns, we now include a new Figure S1 showing secretion of BteA by both wild-type and complemented strains during cultivation in Stainer-Scholte medium, and strain cytotoxicity towards HeLa cells. These results confirm that the T3SS is functional in *B. pertussis* B1917 isolate and support the use of the *BpΔbteA* and complemented strains in our experiments. The strains showed T3SS activity at the time they were added to the hNEC cells. Moreover, several proteins of T3SS injectisome were detected at 48 h post-infection, Table S10. However, we were not able to monitor T3SS activity during the course of hNEC infection. This is a challenging task as T3SS activity can be regulated at multiple levels, transcriptional, translational and even post-translational. We hope to investigate these regulations in our future studies.

Regarding the complemented strain shown in Figure 5, we chose to keep the original dataset as it represents the entire data obtained during the study. However, we have now explicitly mention concerns about the suitability of the complemented strain (highlighted lines 273-276, 308-309, and 325-327).

4. The information presented in Figure 6S regarding the different adhesion sites of the reference strain Tohama I and the clinical isolate B1914 is highly significant, as it appears to suggest that modes of adhesion to nasal epithelial cells may have evolved over time. This finding could have important implications for understanding the adaptations this pathogen has undergone during its continued circulation in the population. It may represent one of the important results of this study and should be included in the main body of the manuscript.

A4: We thank the reviewer for this suggestion. As a result, we have moved the figure from the supplementary material to the main manuscript (now Figure 6).

Reviewer #2:

General comment:

This paper by Zmuda et al describe the responses of Bordetella pertussis to nasal epithelial cells in a primary cell culture. Pertussis disease is still causing a significant amount of deaths in infant under 5 years of age. Importantly, subclinical infections with B. pertussis allowing for constant circulation within the population. The authors develop a primary nasal epithelial cell culture model, that allows for the investigation of first responses from host as well as first adaptations of the bacterium to the host environment. This is a well written manuscript that present the development of a new in vitro model to investigate first responses of human nasal epithelia to Bordetella pertussis as well as the adaptation of Bordetella pertussis to human nasal epithelia. This manuscript validates the model which presents itself very useful, while the authors also state the limitations that need to be consider when using the model.

Comments:

1. The authors discuss that ACT mutant might contribute (Line 370) to barrier disruption. Adding a deletion mutant and complemented strain to confirm this hypothesis should be included in the barrier disruption experiments.

A1: We appreciate the suggestion but decided not to include the ACT-deficient and complemented strains in this study as our primary objective was to investigate the role of the T3SS effector BteA. We point out in the discussion that future work should investigate the role of ACT in barrier modulation of hNECs, and confirm this hypothesis (highlighted lines 381-384).

2. CRT2 downregulated in the BteA mutant and restored in the complemented strain. CRT2 can act as regulator of Ca signaling. The authors have previously described that BteA disrupt calcium homeostasis. A discussion is needed as this might be relevant to further understand the role of BteA as well as can serve as foundation for future investigations.

A2: We thank the reviewer for highlighting this potential connection. The Discussion now includes lines 478-488 linking CRT2 to calcium-dependent signaling pathways and also refers to our previous findings of BteA-mediated disruption of calcium homeostasis.

3. BAZ1B and PHLDB2 are also restored upon BteA complementation in the proteomic analysis while they are also not discussed. BAZ1B discussion might not be relevant at this point due to its reported function up to date. However, the role of PHLDB2 is to regulate EGFR which is an epidermal growth factor, and it can be relevant in the context of the model employ in these experiments.

A3: We have revised the discussion to mention PHLDB2, and highlight its role in regulation of cytoskeleton organization and EGFR signaling. We also briefly mention BAZ1B (lines 488 – 491).

4. Finally, more references are needed specially in discussion and methods to be able to go to original sources

A4: We have revised the manuscript to describe the methods more thoroughly and to include additional references in the Method and Discussion sections that direct the reader to the original sources. We have also included additional tables S20-S21 that specify the compositions of the used media.

Re: Spectrum01267-25R1 (**Limited response of primary nasal epithelial cells to *Bordetella pertussis* infection**)

Dear Dr. Jana Kamanova:

Your manuscript has been accepted, and I am forwarding it to the ASM production staff for publication. Your paper will first be checked to make sure all elements meet the technical requirements. ASM staff will contact you if anything needs to be revised before copyediting and production can begin. Otherwise, you will be notified when your proofs are ready to be viewed.

Sincerely,
Catherine Brissette
Editor
Microbiology Spectrum